# Cerebral Circulation and Brain Temperature during an Ultra-Short Session of Dry Immersion in Young Subjects

**Liudmila Gerasimova-Meigal** *[ID], **Alexander Meigal** [ID], **Maria Gerasimova, Anna Sklyarova** and **Ekaterina Sirotinina**

Department of Physiology and Pathophysiology, Petrozavodsk State University, 33, Lenin Pr., 185910 Petrozavodsk, Russia; meigal@petrsu.ru (A.M.)
* Correspondence: gerasimova@petrsu.ru; Tel.: +7-911-402-9907

**Abstract:** The primary aim of the study was to assess cerebral circulation in healthy young subjects during an ultra-short (45 min) session of ground-based microgravity modeled by "dry" immersion (DI), with the help of a multifunctional Laser Doppler Flowmetry (LDF) analyzer. In addition, we tested a hypothesis that cerebral temperature would grow during a DI session. The supraorbital area of the forehead and forearm area were tested before, within, and after a DI session. Average perfusion, five oscillation ranges of the LDF spectrum, and brain temperature were assessed. Within a DI session, in the supraorbital area most of LDF parameters remained unchanged except for a 30% increase in respiratory associated (venular) rhythm. The temperature of the supraorbital area increased by up to 38.5 °C within the DI session. In the forearm area, the average value of perfusion and its nutritive component increased, presumably due to thermoregulation. In conclusion, the results suggest that a 45 min DI session does not exert a substantial effect on cerebral blood perfusion and systemic hemodynamics in young healthy subjects. Moderate signs of venous stasis were observed, and brain temperature increased during a DI session. These findings must be thoroughly validated in future studies because elevated brain temperature during a DI session can contribute to some reactions to DI.

**Keywords:** microgravity; "dry" immersion; weightlessness; Laser Doppler Flowmetry (LDF); blood perfusion; cerebral circulation; temperature





## 1. Introduction

Several ground-based models of microgravity are currently widely used to mimic the effects of real weightlessness, e.g., supine bed rest, free-fall machines, unilateral lower limb suspension, parabolic flights, and "wet" and "dry" immersion (DI) [1–3]. Among these models, the condition of DI is regarded as one of the most relevant and close to the condition of real weightlessness [4]. DI stands for immersion in fresh warm water without direct contact with water, and it is induced by the conditions of supportlessness, uniform compression of the body, and hypokinesia [4]. In turn, these physical effects provoke such physiological phenomena as redistribution of body fluids in bottom-up direction and "deafferentation". Either real microgravity or its ground-based models exert a strong effect on the physiological systems of a human organism, primarily on the musculoskeletal, nervous, and cardiovascular systems [4–7].

Typically, longer-term sessions of DI (from 3 days to 2–3 weeks) are chosen to study the effect of ground-based microgravity on the organism [8–10] because they generally correspond with the time period of a short-term space flight. Several studies are found in the field of the early (hours) effects of DI on the organism [11,12]. Only a few studies present data on the effect of an ultra-short DI session (less than one hour) on the cardiovascular [13,14] or neuromuscular systems [15].

Sessions of microgravity, which last for hours, days, and weeks, either onboard a space vehicle or modeled, induce an increase in intracranial pressure, increase of jugular vein flow,

and modification in cerebrovascular autoregulation [1,4,12,16–19]. The cerebral circulation during ultra-short DI sessions (less than one hour) has not yet been assessed. Based on the aforementioned studies, one can suppose that some initial signs of modification in cerebral perfusion could well have taken place even within the first hour of DI. It seems important to study cerebral circulation in such shorter DI sessions because shorter space flights (within one hour) are increasingly planned and even taking place due to space tourism industry development. In addition, ultra-short DI sessions are already applied for rehabilitation purposes in neurology and sport medicine [4,13–15].

Several studies have shown that fluid shifts to the upper part of the body during a DI session [2–4]. Theoretically, such a shift must have led to the increase in cerebral temperature. There are several reasons for brain temperature to increase in healthy and diseased conditions [20]. Several studies have shown that hyperthermia, either during exercise-induced heat stress or passive heating, exerts an effect on the neuromuscular and cardiovascular systems [21–26]. For example, heat stress can reduce central nervous activation of muscles and impair afferent feedback from them [21]. These effects are consistent with the concept of "deafferentation" during microgravity and, thus, they can directly contribute to the decrease of muscle tone, which is readily seen under the conditions of DI or spaceflight. From the thermoregulatory point of view, the decrease in muscle tone during hyperthermia is beneficial as it helps blood pumping to the skin vasculature for further heat loss, rather than to skeletal muscles [21,22]. In our earlier study [13] we argued that the condition of DI at 32 °C cannot be regarded as an efficient thermal stimulus. However, brain temperature was not measured in those studies. Altogether, we hypothesize that brain temperature may still have increased during a DI session and it would have contributed to the hypotonic neuromuscular effect of microgravity.

On the whole, the primary aim of the study was to assess cerebral circulation in healthy young human subjects during an ultra-short DI session. In addition, we aimed to test the hypothesis that cerebral temperature would grow during an ultra-short DI session. Laser Doppler Flowmetry (LDF) is acknowledged as a reliable method to assess local cerebral blood flow [27]. In addition, LDF allows the non-invasive measurement and evaluation of cerebral perfusion, which is essential in the condition of DI. Therefore, in this study we measured cerebral blood flow and brain temperature with LDF in young healthy subjects during a single 45 min DI session.

## 2. Materials and Methods

### 2.1. Participants

The study group was formed in accordance with previously published criteria for safe use under the conditions of DI [13,14]. Volunteers were invited to participate in the study, and the purpose of the study and the methods used were explained to them. The subjects who accepted the invitation were then checked for the exclusion criteria (chronic diseases including, for example, arterial hyper- or hypotension, thrombophlebitis, acute inflammatory diseases, a history of traumatic brain injury, and cardiac arrhythmia) [13–15]. In addition, some basic examinations were conducted (body temperature and blood pressure (BP), heart rate (HR), body weight, height). Participants were selected based on the uniformity of their body mass index (BMI) to exclude overweight and older participants.

The study enrolled 11 apparently healthy individuals aged 18–31 years old (mean age 20.7 ± 3.5 years) without chronic diseases. The anthropometric characteristics of participants are presented in Table 1.

None of the subjects had a history of traumatic brain injuries, including those related to sports activities such as boxing and football. An exhaustive verbal explanation was provided to all participants, and written informed consent was obtained from each of them. The protocol of the study was approved by the Joint Ethics Committee of the Ministry of Health of the Republic of Karelia and Petrozavodsk State University (Statement of Approval No. 31, 18 December 2014).

**Table 1.** Anthropometric characteristics of the participants at the time of their inclusion in the study.

| Parameter | Men (n = 4) | Women (n = 7) |
|---|---|---|
| Body Mass, kg | 72.3 ± 3.4 | 57.9 ± 6.4 |
| Height, m | 1.81 ± 0.03 | 1.67 ± 0.05 |
| BMI [1] | 22.1 ± 0.7 | 20.7 ± 2.7 |

[1] BMI, body mass index.

Prior the DI session, all subjects underwent an active orthostatic tolerance test [13,14,28]. The cardiovascular reactivity during this test indicates the baroreflex sensitivity [13,14,28]. To evaluate the active orthostatic test, BP and HR were first measured at the 15th min in subjects lying supine. After that, BP and HR were measured in subjects in a standing position, on the 5th min of standing. HR increased by 4–10 beats per minute, and diastolic BP was slightly (4–6 mm Hg) increased. This implies that none of the subjects had orthostatic hypotension. All subjects were instructed to avoid exhaustive physical activity, alcohol consumption, and nights shifts the day prior to the study.

*2.2. The DI Session*

The condition of DI was induced with help of the "Medical Facility of Artificial Weightlessness" (MEDSIM, Center for Aerospace Medicine and Technologies, State Scientific Center of Russian Federation "Institute of Biomedical Problems," Moscow, Russia), which is located at Petrozavodsk State University. The DI procedure was previously described in detail in our earlier papers [13–15]. MEDSIM is a 2 m$^3$ bath filled with fresh water. The bath was covered with a thin waterproof film of a large size, which allowed for the body of a subject to be wrapped. Additionally, each subject was wrapped in an individual cotton sheet for more comfort, and to avoid direct contact with rubber material of the film. To provide safety and instant monitoring of subject's condition, at least two investigators were present in the laboratory room during the whole DI procedure.

To conduct the DI procedure, the subject was immersed in the water using a movable platform, which was driven by an electric motor. The subject was immersed with their head and upper body "out of water". The subject's arms were freely lying non-flexed, to provide circulation. More information about the physics and the DI procedure can be found in the previous studies [2,4,8,14]. At the end of the DI procedure, the movable platform raised the subject back to a position above water level.

The room temperature was 23–24 °C and the water temperature in the bath was set at 32 °C. Prior to the procedure, the subjects were allowed to drink 200 mL of fresh water and visit the toilet, as DI has strong diuretic effect. Then, the subjects, wrapped for convenience in a cotton sheet, laid supine on the platform for 15 min to familiarize themselves with the conditions of the experiment. BP was measured at the 10th minute of the preparatory period (UA-767, A&D Company Ltd., Toshima City, Japan). The beginning of the DI procedure was approved if the BP was no higher than 140/95 mmHg. Then, the subjects were immersed in water for 45 min, with the possibility to abandon the procedure at their request.

*2.3. Outcome Measures*

Data were collected at the following study points: before (baseline test—preDI), on the 15th, 30th, 40th min of DI session (15′DI, 30′DI and 40′DI, correspondingly), and 3 min after DI (postDI). Systolic and diastolic BP and HR were measured at these study points. ECG was monitored in standard lead II with a "VNS-Spektr" device (Neurosoft Ltd., Ivanovo, Russia).

To evaluate microcirculation, LDF measurements were conducted with the help of a multifunctional portable laser blood microcirculation analyzer "LASMA PF" (SPE LAZMA Ltd., Moscow, Russian Federation, http://www.lazma.ru (accessed on 30 March 2023)) [29,30]. For LDF measurements, four separate identical telemetric devices, "LASMA PF", were used. Two devices were fixed on the forehead on the left and right supraor-

bital regions over the brow skin areas, corresponding to a branch of the internal carotid (ophthalmic) arteries, which supplies blood to the supraorbital artery from the internal carotid artery (Figure 1) [31–33]. Two other devices were fixed on the outer surface of the forearm 5 cm proximal to the wrist. LDF signals were measured up to 300 s with time step 1t = 0.05 s. These devices allowed the recording of three kinds of signals: blood perfusion, skin temperature, and motion. The system was supplied with a wireless data acquisition module.

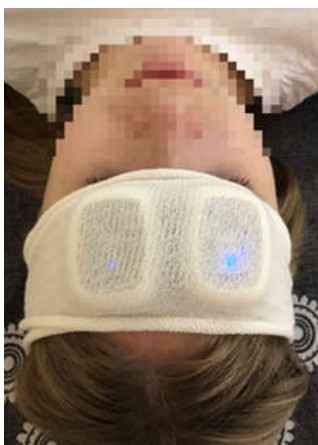

**Figure 1.** Device "LASMA PF" placement for LFD measurements of supraorbital regions.

After data acquisition, the rhythms of oscillations were computed using the built-in wavelet analysis module. Microcirculation was characterized by the LDF signal in arbitrary perfusion units (p.u.), using the software supplied with the LDF analyzer. Mathematical processing of the results included evaluation of the following parameters for each fragment: the average value of perfusion (M), the standard deviation ($\sigma$), and the coefficient of variation KV = ($\sigma$/M)100%. Nutritive ($M_{NUTR}$) and shunt ($M_{SHUNT}$) components of perfusion were also obtained [34]. In addition, the maximal amplitude of oscillations in five frequency domains was computed for assessing the main mechanisms affecting microcirculation [29,32,35,36]: the endothelial activity, 0.0095–0.021 Hz (AE); neurogenic activity, 0.021–0.052 Hz (AN); myogenic activity of vascular smooth muscles, 0.052–0.145 Hz (AM); respiratory-associated rhythm (venular), 0.145–0.6 Hz (AR); and cardiac activity, 0.6–2 Hz (AC). Data obtained from the right and left side of the studied areas were averaged.

### *2.4. Statistical Analysis*

Data were analyzed using IBM SPSS Statistics 21.0 software (SPSS, IBM Company, Chicago, IL, USA). Within the DI session, ANOVA, followed by post-hoc comparisons (Duncan test), was applied to compare LDF and hemodynamic parameters among study points. The significance was considered at $p < 0.05$.

### **3. Results**

### *3.1. Hemodynamics during a DI Session*

At baseline condition (preDI), systolic and diastolic BP was 94–116 and 58–72 mm Hg, respectively, and HR was 59–72 $min^{-1}$ (Figure 2). During the DI session, BP and HR values remained stable. Nevertheless, a slight decrease in both systolic and diastolic BP by 4–5 mm Hg was recorded. The hemodynamic data obtained in men and women did not differ from each other, and therefore were combined them into one common study group.

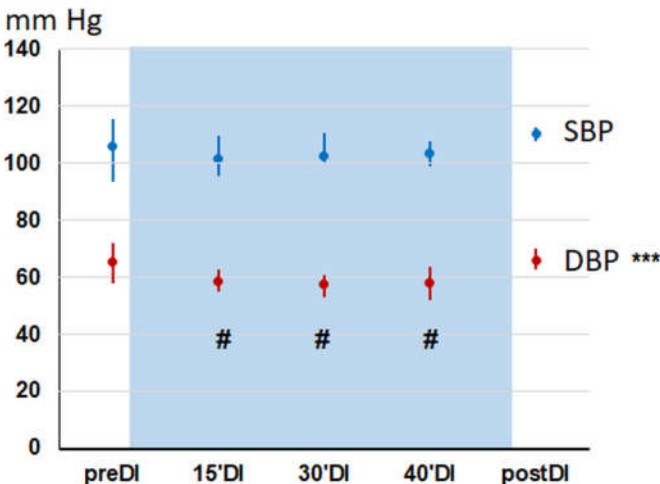

**Figure 2.** Blood pressure in subjects during "dry" immersion session (median and interquartile range). Study points: before (baseline test—preDI); on the 15, 30, 40 min of DI session (15′DI, 30′DI and 40′DI, correspondingly); and 3 min after DI (post-DI). SBP, systolic blood pressure; DBP, diastolic blood pressure. The significance of ANOVA *** $p < 0.001$; the difference from preDI value # $p < 0.001$ (Duncan post-hoc test).

### 3.2. LDF Measurements in the Forehead Supraorbital Area during a DI Session

The parameters of LDF measurements in the forehead area are presented in Table 2. In the forehead area in preDI condition, the average value of perfusion (M) was in the range of 12–17 p.u. During the DI session, there was a tendency towards an increase of the average value of perfusion, though it was not significant. The nutritive and shunt components of perfusion ($M_{NUTR}$ and $M_{SHUNT}$), as well as the overall perfusion variability ($\sigma$ and KV), corresponded with regulatory mechanisms of perfusion and remained stable.

**Table 2.** LDF parameters of the supraorbital area in subjects during a DI session (Mean $\pm$ SD).

| Parameter | PreDI | 15′DI | 30′DI | 30′DI | PostDI | Significance [1] |
|---|---|---|---|---|---|---|
| M, p.u. | $15.14 \pm 4.58$ | $18.48 \pm 4.22$ | $18.45 \pm 3.95$ * | $19.01 \pm 4.68$ | $18.45 \pm 4.59$ * | 0.064 |
| $M_{NUTR}$, p.u. | $6.66 \pm 3.31$ | $8.66 \pm 3.67$ | $7.74 \pm 3.67$ | $9.07 \pm 3.28$ | $9.48 \pm 4.33$ | 0.153 |
| $M_{SHUNT}$, p.u. | $8.48 \pm 3.95$ | $9.81 \pm 4.30$ | $10.71 \pm 3.42$ | $9.95 \pm 2.85$ | $9.00 \pm 4.64$ | 0.440 |
| $\sigma$, p.u. | $1.55 \pm 0.455$ | $1.59 \pm 0.42$ | $1.69 \pm 0.32$ | $1.53 \pm 0.28$ | $1.61 \pm 0.42$ | 0.703 |
| KV | $10.68 \pm 3.23$ | $8.87 \pm 2.61$ | $9.47 \pm 2.14$ | $8.55 \pm 2.76$ | $8.96 \pm 2.12$ | 0.117 |
| AE, $W^2$ | $0.43 \pm 0.17$ | $0.36 \pm 0.30$ | $0.41 \pm 0.18$ | $0.31 \pm 0.14$ | $0.37 \pm 0.21$ | 0.475 |
| AN, $W^2$ | $0.47 \pm 0.15$ | $0.40 \pm 0.22$ | $0.59 \pm 0.23$ | $0.47 \pm 0.18$ | $0.48 \pm 0.18$ | 0.048 |
| AM, $W^2$ | $0.54 \pm 0.22$ | $0.66 \pm 0.31$ | $0.65 \pm 0.22$ | $0.69 \pm 0.21$ | $0.76 \pm 0.32$ | 0.177 |
| AR, $W^2$ | $0.34 \pm 0.11$ | $0.48 \pm 0.17$ * | $0.48 \pm 0.13$ * | $0.47 \pm 0.14$ ** | $0.51 \pm 0.19$ ** | 0.008 |
| AC, $W^2$ | $0.82 \pm 0.30$ | $1.023 \pm 0.29$ | $1.06 \pm 0.26$ | $1.02 \pm 0.25$ | $1.06 \pm 0.36$ | 0.075 |
| T, °C | $34.84 \pm 1.04$ | $38.05 \pm 0.31$ *** | $38.40 \pm 0.26$ *** | $38.45 \pm 0.32$ *** | $38.36 \pm 0.35$ *** | 0.000 |

[1] Study points: before (baseline test—preDI); on the 15, 30, 40 min of DI session (15′DI, 30′DI and 40′DI, correspondingly); and 3 min after DI (post-DI). M, average value of perfusion; $M_{NUTR}$, average value of nutritive perfusion; $M_{SHUNT}$, average value of shunt perfusion; $\sigma$, standard deviation; KV, coefficient of variation; AE, endothelial activity; AN, neurogenic activity; AM, myogenic activity; AR, respiratory associated activity; AC, cardiac activity; T, temperature. The significance is based on ANOVA with further post-hoc comparisons (Duncan test); the difference from the baseline condition: * $p < 0.05$, ** $p < 0.01$, *** $p < 0.001$.

Frequency domain analysis revealed a significant increase in respiratory modulated activity (AR), which corresponds to oscillations in the venous vascular bed. In addition, minor changes in the neurogenic (AN) and cardiac (AC) LDF frequency ranges were revealed.

### 3.3. LDF Measurements in the Forearm Area during a DI Session

Values of LDF parameters of the forearm area are presented in Table 3. In preDI condition, the average value of perfusion (M) was in the range of 3–7 p.u. During the

DI session, there was a significant increase in the average value of perfusion (M) and nutritive component of perfusion ($M_{NUTR}$). The shunt component of perfusion ($M_{SHUNT}$) remained unchanged. The overall perfusion variability (σ) slightly increased, while KV did not change.

**Table 3.** LDF parameters of the forearm area in subjects during a DI session (Mean ± SD).

| Parameter | PreDI | 15'DI | 30'DI | 30'DI | PostDI | Significance [1] |
|---|---|---|---|---|---|---|
| M, p.u. | 4.46 ± 1.08 | 6.94 ± 2.71 | 8.56 ± 4.48 ** | 9.50 ± 5.33 ** | 8.69 ± 3.35 * | 0.005 |
| $M_{NUTR}$, p.u. | 2.39 ± 1.71 | 3.47 ± 1.69 | 5.10 ± 4.30 * | 5.41 ± 3.84 | 5.23 ± 2.71 | 0.038 |
| $M_{SHUNT}$, p.u. | 1.31 ± 0.83 | 2.68 ± 0.94 | 3.01 ± 1.91 | 3.38 ± 1.97 | 3.04 ± 1.57 | 0.241 |
| σ, p.u. | 0.43 ± 0.10 | 0.59 ± 0.36 | 0.84 ± 0.57 * | 0.76 ± 0.46 | 0.70 ± 0.36 | 0.067 |
| KV | 9.78 ± 2.13 | 8.32 ± 1.66 | 9.71 ± 3.21 | 8.24 ± 3.49 | 8.07 ± 2.00 | 0.205 |
| AE, $W^2$ | 0.16 ± 0.15 | 0.14 ± 0.11 | 0.19 ± 0.12 | 0.15 ± 0.14 | 0.17 ± 0.09 | 0.789 |
| AN, $W^2$ | 0.18 ± 0.11 | 0.22 ± 0.17 | 0.25 ± 0.18 | 0.26 ± 0.22 | 0.19 ± 0.09 | 0.593 |
| AM, $W^2$ | 0.20 ± 0.09 | 0.26 ± 0.13 | 0.39 ± 0.38 | 0.36 ± 0.26 | 0.34 ± 0.19 | 0.178 |
| AR, $W^2$ | 0.13 ± 0.06 | 0.17 ± 0.08 | 0.22 ± 0.12 * | 0.23 ± 0.12 | 0.22 ± 0.10 | 0.047 |
| AC, $W^2$ | 0.20 ± 0.05 | 0.34 ± 0.20 | 0.40 ± 0.26 | 0.38 ± 0.21 * | 0.39 ± 0.25 | 0.067 |
| T, °C | 32.74 ± 1.12 | 35.75 ± 1.25 *** | 36.36 ± 1.18 *** | 36.44 ± 1.17 *** | 36.42 ± 1.18 *** | 0.000 |

[1] Study points: before (baseline test—preDI); on the 15, 30, 40 min of DI session (15'DI, 30'DI and 40'DI, correspondingly); and 3 min after DI (post-DI). M, average value of perfusion; $M_{NUTR}$, average value of nutritive perfusion; $M_{SHUNT}$, average value of shunt perfusion; σ, standard deviation; KV, coefficient of variation; AE, endothelial activity; AN, neurogenic activity; AM, myogenic activity; AR, respiratory associated activity; AC, cardiac activity; T, temperature. The significance is based on ANOVA with further post-hoc comparisons (Duncan test); the difference from the baseline condition: * $p < 0.05$, ** $p < 0.01$, *** $p < 0.001$.

Frequency domain analysis revealed a significant increase in respiratory modulated (AR) and cardiac (AC) activity.

### 3.4. Temperature Measurements during a DI Session

The temperature dynamics in the studied areas is shown in Figure 3 and mean values are presented in Tables 2 and 3. During a DI session, in the suborbital area the temperature varied within a narrow range of 36.8–38.5 °C, while in the forearm area it varied over a wider range of 34.2–38.0 °C.

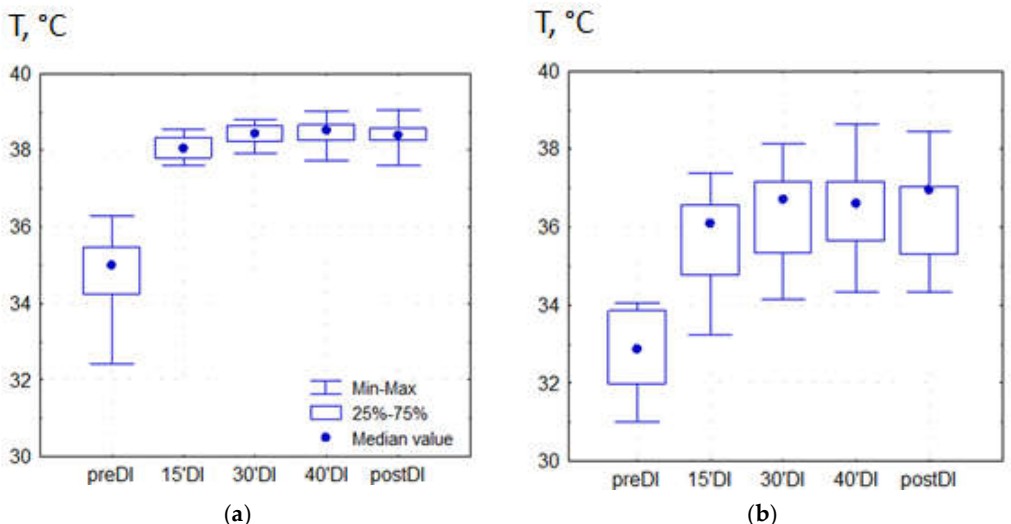

**Figure 3.** The temperature of the suborbital area (**a**) and forearm area (**b**).

## 4. Discussion

Primarily, in this study we aimed to evaluate cerebral circulation in healthy young human subjects during an ultra-short 45 min DI session. In addition, we tested a hypothesis that cerebral temperature would grow during an ultra-short DI session in heathy subjects.

It was found that only a few parameters of the LDF signal had indeed modified within a 45 min session of DI, while others had not. More specifically, the amplitude of the respiratory (AR) component of the LDF spectrum had significantly increased by roughly 30% in comparison to the pre-DI condition. The myogenic (AM), endothelial (AE), and cardiac (AC) components of the LDF spectrum in the supraorbital area did not change within the DI session. The mean value of perfusion (M) increased from 15 p.u. before DI up to 19 p.u. by the end of the DI session, though non-significantly. It has been found that the temperature, measured in the supraorbital area, indeed increased during the DI session. As for the forearm area, its perfusion was initially significantly lower by three times and it significantly increased within a DI session. Furthermore, in the forearm area nutritive perfusion increased during DI. Thus, there was a distinct difference in the reactivity of the two studied areas.

The dynamics of BP in young healthy individuals is consistent with our earlier results [13,14] as well as data from other studies [16,37], which reported a 5 mm decrease in diastolic BP associated with a 7% decrease in total peripheral vascular resistance. This indicates a high reproducibility of the effect of a short DI session on hemodynamics.

The data obtained indicate, in general, the stability of the cerebral blood flow during a DI session because most of LDF parameters did not change. Oscillations in the respiration dependent frequency range predominantly indicate the venous blood flow because the respiratory system is one of the major drivers for venous return to the heart [38,39]. The respiratory component of LDF is believed to increase during venous stasis [34,35]. Therefore, as this component of the LDF spectrum increased, one can conclude that minor venous stasis still took place during a 45 min DI session. These findings largely correspond with previous studies, which evidenced venous stasis and growth of intracranial pressure in subjects during longer-term DI sessions (3 to 5 days) [4,18,19]. As for the changes in the neurogenic dependent frequency range of the LDF signal, they were non-significant in the forearm area and were on the border of significance in the forehead area. This suggests that the neurogenic dependent oscillations of the LDF signal were relatively stable during 45 min DI session.

In the forearm area, unlike the forehead area, the most notable change was the increase of the average value of perfusion (M) and nutritive component of perfusion ($M_{NUTR}$). Such a big, almost double, increase in perfusion corresponds with hyperemia, which in turn may be induced for the purpose of thermoregulation, more specifically, for the purpose of heat loss [40]. This assumption is supported by the growth of hand skin temperature during the DI session.

Brain temperature in vivo cannot be directly measured without invasive probes. Most direct measurements of brain temperature were obtained during neurosurgical operations [41]. Therefore, most of the information on brain temperature was obtained with the help of indirect methods, e.g., its computation based on whole-brain magnetic resonance spectroscopy (MRS) [41]. The MRS method of temperature calculation is acknowledged as reliable [42]. Even so, this method is science-intensive and expansive. In addition, this method is not suitable for the monitoring of brain temperature in subjects positioned in a special environment or device (DI bathtub). Brain temperature can be noninvasively measured with an infrared thermography of the eyeballs [43], which reliably indicates the central body temperature. However, this method is also indirect and it is not yet clear whether it measures the temperature of the retina or cornea. Therefore, since the LDF device used in our study provided thermal data during the DI session, we used it as the most reliable method of temperature monitoring.

Brain temperature is usually higher than that of the "core" and it ranges in healthy subjects from 36.1 to 40.9 °C (mean brain temperature 38.5 ± 0.4 °C), depending on age, time of day, and hormonal status [41]. In the present study, in normal healthy subjects the temperature grew within a 45 min DI session in the supraorbital area, and lay within normal range. In all subjects, the dynamics of supraorbital skin temperature was largely the same, which indicates the high reproducibility of the effect of DI. We regard this finding as

important because brain temperature as a part of the thermal "core" of the organism must be subjected to thermal homeostasis. Growth of brain temperature during DI, therefore, can theoretically contribute to some well-known effects of DI, for example, skeletal muscle hypotonia. The interactive effects of temperature and gravity on muscle tone is currently under discussion in academic community [26]. Furthermore, we aware of the fact that supraorbital temperature strongly relates to brain temperature, but it is not necessarily identical with it. We assume that this finding must be validated in further studies with help of diverse methods of temperature measurement.

Several limitations to the study can be identified. First, brain temperature was evaluated with help of an LDF device in the supraorbital area of forehead. An MRS-based measurement of brain temperature would be more relevant. Due to necessity of constantly monitoring brain temperature, the LDF-based method was chosen. In addition, we did not assess whether the horizontal position per se contributed to the outcome. In our future studies, we plan to elucidate this issue. Finally, the temperature of deep and superficial areas of the brain varies, which was not considered in the study.

## 5. Conclusions

In conclusion, our results suggest that an ultra-short, 45 min long, DI session does not exert substantial effects on cerebral blood perfusion in young healthy subjects. Systemic hemodynamics seen from BP were also stable. Nonetheless, some moderate signs of venous stasis of cerebral perfusion were observed.

In addition, it was found, in accordance with the working hypothesis, that brain temperature indeed increased in the course of a 45 min DI session. This finding must be thoroughly validated in future studies because elevated brain temperature during a DI session could well have contributed to some typical reactions to DI, e.g., decreased muscle tone.

These findings can be additionally tested with help of other models of Earth-based microgravity, e.g., during bed rest.

**Author Contributions:** Conceptualization, L.G.-M. and A.M.; methodology, L.G.-M. and A.M.; validation, L.G.-M. and A.M.; formal analysis, L.G.-M., M.G., A.S. and E.S.; investigation, L.G.-M., M.G., A.S. and E.S.; resources, A.M.; data curation, L.G.-M., M.G., A.S. and E.S.; writing—original draft preparation, L.G.-M. and A.M.; writing—review and editing, L.G.-M. and A.M.; visualization, L.G.-M.; supervision, L.G.-M.; project administration, A.M.; funding acquisition, A.M. All authors have read and agreed to the published version of the manuscript.

**Funding:** This research received no external funding.

**Institutional Review Board Statement:** The study was conducted in accordance with the Declaration of Helsinki and approved by the joint Ethics Committee of the Ministry of Health care of the Republic of Karelia and Petrozavodsk State University (Statement of approval No. 31, 18 December 2014).

**Informed Consent Statement:** Informed consent was obtained from all subjects involved in the study.

**Data Availability Statement:** Not applicable.

**Acknowledgments:** The authors thank the subjects for their participation.

**Conflicts of Interest:** The authors declare no conflict of interest.

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
