# Peer review of "Cerebral Circulation and Brain Temperature during an Ultra-Short Session of Dry Immersion in Young Subjects"

_pathophysiology, doi:10.3390/pathophysiology30020018_

Round 1
Reviewer 1 Report
One of my concerns is the date of the ethical approval. It belongs to 2014 (nine years ago). As far as I know ethical approval must be obtained for each study specifically. I am wondering if this research started 9 years ago. Maybe details of ethical approval should be asked.
The other issue is the justification of methodology for measuring the thermal information. There are some other methods such as in Magnetic Resonance Image Guided Focused ultrasound, where MR is utilized for obtaining thermal information. However MR is an expensive thermometer and they could mention this for their selection by referring to the below suggested references. However it would be interesting to obtain a regional thermal map of the brain etc.
Maybe authors could mention to the below references and provide pros and cons while comparing.
Senay Mihcin, Ioannis Karakitsios, Nhan Le, Jan Strehlow, Daniel Demedts, Michael Schwenke, Sabrina Haase, Tobias Preusser, Andreas Melzer, Methodology on quantification of sonication duration for safe application of MR guided focused ultrasound for liver tumour ablation,Computer Methods and Programs in Biomedicine,Volume 152,2017,Pages 125-130,ISSN 0169-2607,https://doi.org/10.1016/j.cmpb.2017.09.006.
Schwenke, M., Strehlow, J., Demedts, D. et al. A focused ultrasound treatment system for moving targets (part I): generic system design and in-silico first-stage evaluation. J Ther Ultrasound 5, 20 (2017). https://doi.org/10.1186/s40349-017-0098-7
The use of heat in tumor ablation is well known as it causes coagulation. Authors suggest in their conclusion, validation, and adverse effects, I think they should make sure that they are not causing harm in their subjects.
Also SWOT analysis of the study in the conclusion part could be useful. Strengths of the utilized methods, weakness, and opportunities, and limitations of the study.
Author Response
Reviewer 1
Q1: One of my concerns is the date of the ethical approval. It belongs to 2014 (nine years ago). As far as I know ethical approval must be obtained for each study specifically. I am wondering if this research started 9 years ago. Maybe details of ethical approval should be asked.
Answer
Indeed, we used this protocol for 9 years. It still totally corresponds to the topic of the study and justifies it. No changes or modifications were added to the protocol because the method of "dry" immersion is a standard regular model of Earth-based microgravity officially adopted for space-related studies in Russia (and not only in Russia, also in France). We have a brochure and a protocol of safety for this method. Several papers were published within the years 2021-2023 supported by this protocol. The list of papers is presented below (including those published by MDPI).
Still, we would agree that the ethical approval looks outdated. Three weeks ago we applied for a new version of this protocol. However, even with new version, we should have used the previous one, because the measurements were conducted several months earlier. As such we would like to preserve the older version of the ethical approval in the text.
- Meigal AY, Tretjakova OG, Gerasimova-Meigal LI, Sayenko IV. Program of Seven 45-min Dry Immersion Sessions Improves Choice Reaction Time in Parkinson's Disease. Front Physiol. 2021, 11:621198. doi: 10.3389/fphys.2020.621198.
- Gerasimova-Meigal L, Meigal A, Sireneva N, Saenko I. Autonomic Function in Parkinson's Disease Subjects Across Repeated Short-Term Dry Immersion: Evidence From Linear and Non-linear HRV Parameters. Front Physiol. 2021, 12:712365. doi: 10.3389/fphys.2021.712365.
- Meigal AY, Gerasimova-Meigal LI, Reginya SA, Soloviev AV, Moschevikin AP. Gait Characteristics Analyzed with Smartphone IMU Sensors in Subjects with Parkinsonism under the Conditions of "Dry" Immersion. Sensors (Basel). 2022, 22(20):7915. doi: 10.3390/s22207915.
- Gerasimova-Meigal, L.; Meigal, A.; Sireneva, N.; Gerasimova, M.; Sklyarova, A. Heart Rate Variability Parameters to Evaluate Autonomic Functions in Healthy Young Subjects during Short-Term “Dry” Immersion. Physiologia 2023, 3, 119-128. doi.org/10.3390/physiologia3010010
Q2: The other issue is the justification of methodology for measuring the thermal information. There are some other methods such as in Magnetic Resonance Image Guided Focused ultrasound, where MR is utilized for obtaining thermal information. However MR is an expensive thermometer and they could mention this for their selection by referring to the below suggested references. However it would be interesting to obtain a regional thermal map of the brain etc.
Maybe authors could mention to the below references and provide pros and cons while comparing.
Mihcin S, Karakitsios I, Le N, Strehlow J, Demedts D, Schwenke M, Haase S, Preusser T, Melzer A. Methodology on quantification of sonication duration for safe application of MR guided focused ultrasound for liver tumour ablation. Comput Methods Programs Biomed. 2017 Dec;152:125-130. doi: 10.1016/j.cmpb.2017.09.006.
Schwenke M, Strehlow J, Demedts D, Haase S, Barrios Romero D, Rothlübbers S, von Dresky C, Zidowitz S, Georgii J, Mihcin S, Bezzi M, Tanner C, Sat G, Levy Y, Jenne J, Günther M, Melzer A, Preusser T. A focused ultrasound treatment system for moving targets (part I): generic system design and in-silico first-stage evaluation. J Ther Ultrasound. 2017 Jul 24;5:20. doi: 10.1186/s40349-017-0098-7.
Answer
We would agree that MR is a relevant method to measure temperature of the brain. Of course, MR is a science-intensive and expansive method. In addition, the MR method is not reliable for monitoring of the brain temperature, because a subject should be extracted from the experimental set-up (during "dry" immersion). For that reason, we took the opportunity to use built-in thermometry of the LDF device. We agree that justification of thermometry choice is needed and we added it to the text (in the Discussion section). We appreciate reference suggestions. We thoroughly read both references and added one of them to the text.
Q3: The use of heat in tumor ablation is well known as it causes coagulation. Authors suggest in their conclusion, validation, and adverse effects, I think they should make sure that they are not causing harm in their subjects.
Answer
We examined academic literature in this field and found that heat-based tumor ablation is conducted with high intensity focused ultrasound (HIFU) which “utilizes a stereotactic device to distribute high intensity energy (100–10,000 W/cm2) through the skull. This produces spatial ablation at target tumor sites by increasing the temperature to approximately 55oC.” (Arsiwala et al., 2021, see below). In our study, temperature of brain was far from this value.
Arsiwala TA, Sprowls SA, Blethen KE, Adkins CE, Saralkar PA, Fladeland RA, Pentz W, Gabriele A, Kielkowski B, Mehta RI, Wang P, Carpenter JS, Ranjan M, Najib U, Rezai AR, Lockman PR. Ultrasound-mediated disruption of the blood tumor barrier for improved therapeutic delivery. Neoplasia. 2021 Jul;23(7):676-691. doi: 10.1016/j.neo.2021.04.005.
Q4: Also SWOT analysis of the study in the conclusion part could be useful. Strengths of the utilized methods, weakness, and opportunities, and limitations of the study.
Answer:
In the final paragraphs of the Discussion section, we presented a kind of such analysis, including criticism and limitations to the study.
Reviewer 2 Report
The authors of the paper “Cerebral Circulation and Brain Temperature during an Ultra-Short Session of Dry Immersion in Young Subjects “aimed to assess cerebral circulation in healthy young subjects during an ultra-short session of ground-based microgravity modelled by dry immersion with help of multifunctional laser Doppler flowmetry analyzer. Additionally, they tested a hypothesis that cerebral temperature grows during a dry immersion session. The authors suggest that a short-term dry immersion session does not create a substantial effect on cerebral blood perfusion and systemic hemodynamics, however, moderate signs of venous stasis were observed, and brain temperature increased.
1. A native English speaker should check the entire manuscript.
2. Title: The title is adequate.
3. Abstract: The abstract is well-addressed and understandable.
4. Introduction: The introduction is adequately written, however, there are some issues that should be addressed.
It is a little difficult to follow the text due to the switching from one tense to another. For example, line 55- “Theoretically, fluid shift to the upper part of the body during a DI session must have led to the increase in cerebral temperature”- it is unclear whether is it about a work from the other authors or it is addressed to the present study. In the latter case, the results should not be discussed in the introduction.
Line 67: “Those” should be changed.
5. The Materials & Methods: This section is thoroughly and well described.
Line 98: “It was found that none of the subjects had orthostatic hypotension”- should be “None of the subjects had orthostatic hypotension”
Line 110: subjects should be subject
Line 132: Neu-rosoft Ltd should be Neurosoft
6. Results: The results are clearly presented and adequately addressed and the selected articles are recently published.
Figure 3: The caption is moved to another page.
7. Discussion: The discussion is well-written and adequately addressed.
Line 239: there are some extra parentheses.
Overall, the paper “Cerebral Circulation and Brain Temperature during an Ultra-Short Session of Dry Immersion in Young Subjects “is written in an appropriate manner, it is well-structured, technically sound, and above all interesting to read. Due to the attractiveness of the subject, it is likely to attract a wide readership, therefore, the paper can be considered for the publication.
A native English speaker should check the entire manuscript.
Author Response
Reviewer 2
The authors of the paper “Cerebral Circulation and Brain Temperature during an Ultra-Short Session of Dry Immersion in Young Subjects “aimed to assess cerebral circulation in healthy young subjects during an ultra-short session of ground-based microgravity modelled by dry immersion with help of multifunctional laser Doppler flowmetry analyzer. Additionally, they tested a hypothesis that cerebral temperature grows during a dry immersion session. The authors suggest that a short-term dry immersion session does not create a substantial effect on cerebral blood perfusion and systemic hemodynamics, however, moderate signs of venous stasis were observed, and brain temperature increased.
- A native English speaker should check the entire manuscript.
Answer: We have checked English once again and consulted with a professional user of English.
- Title: The title is adequate.
- Abstract: The abstract is well-addressed and understandable.
- Introduction: The introduction is adequately written, however, there are some issues that should be addressed.
It is a little difficult to follow the text due to the switching from one tense to another. For example, line 55- “Theoretically, fluid shift to the upper part of the body during a DI session must have led to the increase in cerebral temperature”- it is unclear whether is it about a work from the other authors or it is addressed to the present study. In the latter case, the results should not be discussed in the introduction.
Answer: We agree with this comment. Fluid shift is a well and long-known phenomenon. We modified the sentence as following:
Several studies have shown that fluid shifts to the upper part of the body during a DI session [2-4]. Theoretically, such a shift must have led to the increase in cerebral temperature.
Line 67: “Those” should be changed.
Answer: Done in accordance.
- The Materials & Methods: This section is thoroughly and well described.
Line 98: “It was found that none of the subjects had orthostatic hypotension”- should be “None of the subjects had orthostatic hypotension”
Answer: According to comments of the other reviewer, we modified the whole paragraph. So this sentence was modified:
“This implies that none of the subjects had orthostatic hypotension”.
Line 110: subjects should be subject
Answer: Done in accordance.
Line 132: Neu-rosoft Ltd should be Neurosoft
Answer: Done in accordance.
- Results: The results are clearly presented and adequately addressed and the selected articles are recently published.
Figure 3: The caption is moved to another page.
Answer: We assume that this issue will be fixed during professional formatting of the text for publishing.
- Discussion: The discussion is well-written and adequately addressed.
Line 239: there are some extra parentheses.
Answer: Done in accordance.
Overall, the paper “Cerebral Circulation and Brain Temperature during an Ultra-Short Session of Dry Immersion in Young Subjects “is written in an appropriate manner, it is well-structured, technically sound, and above all interesting to read. Due to the attractiveness of the subject, it is likely to attract a wide readership, therefore, the paper can be considered for the publication.
Answer: We appreciate so much conceiving the essence of the study.
Reviewer 3 Report
All corrections should be done before the start of publication process
There is no highlights ---should be
There is ethical approval number or code(IACUC)---very old
There is no recommendations
The abstract should be improved and contained background including the aims , Methods, Results and Conclusion
The authors used a huge a mounts of abbreviations---he should tabulate all and be detailed then abbreviate for the ordinary readers
What is the new or novel in this paper did you want to tell ???
LN/25---add cerebral circulation , hypothalamic centers , PGF2 alpha and weightlessness to the keywords
Introduction is very long and the aims of the study not clear
LN/101---what about the head circumference and skin fold thickness?
LN/97-98----Orthostatic tolerance test----for the ordinary readers---clear that
LN/125---is this reference or what ??
LN/127----add reference
LN/157---there is no reference for the statistical analysis
What the relation between this paper and pathology ????
LN/164---figure 1 is very bad
LN/173-177---description should be more summarize
From LN/215-229---this is not discussion
Discussion should be more concise and based upon debating the obtained results with the results of the previous investigators results
Results is very long----should be more summarize
The most used references contained more than 6 authors ---why ??? should be 6 at the maximum plus etal with the last ones ---apply for all
As volume , issue , number and pages ---all are available so no need for the link(s)---apply for all
Some cited references need to be more update
Write like --Table(1):-----------/Fig.(1):----------- apply for all
Okay
Author Response
Reviewer 3
All corrections should be done before the start of publication process
There is no highlights ---should be
Answer: We used the template for preparing the manuscript for this journal. Please, specify what kind of highlights should be added.
There is ethical approval number or code (IACUC)---very old
Answer: Indeed, we used this protocol for 9 years. It still totally corresponds to the topic of the study and justifies it. No changes or modifications were added to the protocol because the method of "dry" immersion is a standard regular model of Earth-based microgravity officially adopted for space-related studies in Russia (and not only in Russia, also in France). We have a brochure and a protocol of safety for this method. Several papers were published within the years 2021-2023 supported by this protocol. The list of papers is presented below (including those published by MDPI).
Still, we would agree that the ethical approval looks outdated. Three weeks ago we applied for a new version of this protocol. However, even with new version, we should have used the previous one, because the measurements were conducted several months earlier. As such we would like to preserve the older version of the ethical approval in the text.
- Meigal AY, Tretjakova OG, Gerasimova-Meigal LI, Sayenko IV. Program of Seven 45-min Dry Immersion Sessions Improves Choice Reaction Time in Parkinson's Disease. Front Physiol. 2021, 11:621198. doi: 10.3389/fphys.2020.621198.
- Gerasimova-Meigal L, Meigal A, Sireneva N, Saenko I. Autonomic Function in Parkinson's Disease Subjects Across Repeated Short-Term Dry Immersion: Evidence From Linear and Non-linear HRV Parameters. Front Physiol. 2021, 12:712365. doi: 10.3389/fphys.2021.712365.
- Meigal AY, Gerasimova-Meigal LI, Reginya SA, Soloviev AV, Moschevikin AP. Gait Characteristics Analyzed with Smartphone IMU Sensors in Subjects with Parkinsonism under the Conditions of "Dry" Immersion. Sensors (Basel). 2022, 22(20):7915. doi: 10.3390/s22207915.
- Gerasimova-Meigal, L.; Meigal, A.; Sireneva, N.; Gerasimova, M.; Sklyarova, A. Heart Rate Variability Parameters to Evaluate Autonomic Functions in Healthy Young Subjects during Short-Term “Dry” Immersion. Physiologia 2023, 3, 119-128. doi.org/10.3390/physiologia3010010
There is no recommendations
Answer: Please, could you specify this comment?
The abstract should be improved and contained background including the aims , Methods, Results and Conclusion.
Answer: We used the template for the journal Pathophysiology, which recommends the following:
“Abstract: A single paragraph of about 200 words maximum. For research articles, abstracts should give a pertinent overview of the work. We strongly encourage authors to use the following style of structured abstracts, but without headings: (1) Background: Place the question addressed in a broad context and highlight the purpose of the study; (2) Methods: briefly describe the main methods or treatments applied; (3) Results: summarize the article’s main findings; (4) Conclusions: indicate the main conclusions or interpretations. The abstract should be an objective representation of the article and it must not contain results that are not presented and substantiated in the main text and should not exaggerate the main conclusions.”
As such, no headings are needed.
The authors used a huge a mounts of abbreviations---he should tabulate all and be detailed then abbreviate for the ordinary readers
Answer: We carefully checked all abbreviations. In the maintext we used only few conventional ones (blood pressure (BP), heart rate (HR), body-mass index (BMI), Laser Doppler Flowmetry (LDF), “dry” immersion (DI)). Abbreviations that were used in tables were explained in the footnote and in the maintext also.
Q: What is the new or novel in this paper did you want to tell ???
Answer: There are novel results in this study. First, brain temperature was for the first time monitored during a ground-based model of microgravity. Second, cerebral flow was evaluated during the first minutes of microgravity, what has not been done earlier.
LN/25---add cerebral circulation , hypothalamic centers , PGF2 alpha and weightlessness to the keywords
Answer: Keywords were modified as microgravity; “dry” immersion; weightlessness; laser Doppler flowmetry (LDF); blood perfusion; cerebral circulation; temperature. Some of the suggested keywords do are not mentioned in the text.
Introduction is very long and the aims of the study not clear
Answer: Two aims have been presented in the Introduction section (last paragraph):
“…the primary aim of the study was to assess cerebral circulation in healthy young human subjects during an ultra-short DI session. In addition, we were aimed at testing a hypothe-sis that cerebral temperature would have grown during an ultra-short DI session. Laser Doppler Flowmetry (LDF) is acknowledged as a reliable method to assess local cerebral blood flow [27]]. In addition, LDF allows to measure and evaluate cerebral perfusion non-invasively, what is essential in the condition of DI. Therefore, in this study we measured cerebral blood flow and brain temperature with LDF in young healthy subjects during a single 45-minute DI session”.
LN/101---what about the head circumference and skin fold thickness?
Answer: We did not measure these parameters. As for skinfolds, we think that BMI (body mass index) adequately reflects this aspect of anthropometry.
LN/97-98----Orthostatic tolerance test----for the ordinary readers---clear that
Answer: We have added information about this test:
“Prior the DI session, all subjects underwent an active orthostatic tolerance test [13,14,28]. The cardiovascular reactivity during this test informs on the baroreflex sensitivity [13,14,28]. To evaluate the active orthostatic test, BP and HR were first measured at the 15th min in subjects lying supine. After that, BP and HR were measured in standing position, on the 5th min of staying. HR increased by 4-10 beats per minute, and diastolic BP was slightly (4-6 mm Hg) increased. This implies that none of the subjects had orthostatic hy-potension. All subjects were instructed to avoid exhaustive physical activity and alcohol consumption and nights shifts the day prior to the study.”
LN/125---is this reference or what ??
Answer: Line 125 does not present any reference. Please, specify.
LN/127----add reference
Answer: The text on line 127 mere describes procedure of dry immersion. No reference is needed.
LN/157---there is no reference for the statistical analysis
Answer: We have indicated the statistical package that was used for the analysis at the end of the Methods section.
What the relation between this paper and pathology ????
Answer: This study finds itself in the field of pathophysiology. It elucidates mechanisms of organism reactivity to novel environment, i.e. ground-based model of microgravity (weightlessness). This has direct link to medical application in the field of rehabilitation. We have published several papers on the use of dry immersion in subjects with Parkinson’s disease. References are listed in the manuscript.
LN/164---figure 1 is very bad
Answer: Could you kindly specify what kind of modifications should be applied to this figure?
LN/173-177---description should be more summarize
Answer: While preparing the legend to Figure 2, we followed “Instructions for Authors”
From LN/215-229---this is not discussion
Answer: This paragraph (lines 215-229) summarizes major outcome of the study in order to facilitate readers to start following discussion.
Discussion should be more concise and based upon debating the obtained results with the results of the previous investigators results
Answer: The Discussion section has been revised and partly re-written.
Results is very long----should be more summarize
Answer: We did our best to present Results as much clearly as possible. The manuscript fits to the recommended count of words (4000 words). As such, we do not consider to make any part of the maintext shorter.
The most used references contained more than 6 authors ---why ??? should be 6 at the maximum plus etal with the last ones ---apply for all
As volume , issue , number and pages ---all are available so no need for the link(s)---apply for all
Some cited references need to be more update
Answer: We followed the template. We will obligatory consult with the Editor on this issue.
Write like --Table(1):-----------/Fig.(1):----------- apply for all
Answer: We followed the template. We will obligatory consult with the Editor on this issue.